# Growth Performance, Antioxidant Capacity, Lipid-Related Transcript Expression and the Economics of Broiler Chickens Fed Different Levels of Rutin

**DOI:** 10.3390/ani9010007

**Published:** 2018-12-22

**Authors:** Fardos A. M. Hassan, Elshimaa M. Roushdy, Asmaa T. Y. Kishawy, Asmaa W. Zaglool, Hammed A. Tukur, Islam M. Saadeldin

**Affiliations:** 1Department of Animal Wealth Development, Faculty of Veterinary Medicine, Zagazig University, Ash Sharqia Governorate 44519, Egypt; fardoseconomy@yahoo.com (F.A.M.H.); shimaa_production@yahoo.com (E.M.R.); asmaa_zaglool@yahoo.com (A.W.Z.); 2Department of Nutrition and Clinical Nutrition, Faculty of Veterinary Medicine, Zagazig University, Ash Sharqia Governorate 44519, Egypt; 3Department of Animal Production, College of Food and Agricultural Science, King Saud University, Riyadh 11451, Saudi Arabia; tukurhammeda@gmail.com; 4Department of Physiology, Faculty of Veterinary Medicine, Zagazig University, Ash Sharqia Governorate 44519, Egypt

**Keywords:** growth performance, antioxidant, lipid expression, broiler, rutin

## Abstract

**Simple Summary:**

Poultry makes a substantial contribution to food security and nutrition. A growing human population and rising incomes have resulted in an increased demand for white meat. Poultry is the fastest growing animal production sector. Rutin, a natural growth and health promoter, was used at three levels for broiler feed (0.25, 0.5 or 1 g rutin/kg). Supplementing broiler diets with rutin, especially at 1 g/kg, has a variety of growth-promoting effects. It enhances antioxidant capacity and suppresses lipogenesis, thereby reducing fat deposition and serum lipid levels. The results demonstrate that the observed benefits can be achieved without compromising economic profits.

**Abstract:**

The effects of rutin on growth performance, hematological and biochemical profiles, antioxidant capacity, economics and the relative expression of selected antioxidants and lipid-related genes were studied in broiler chickens over 42 days. A total of 200 one-day-old female Ross-308 broiler chickens were distributed into four groups, with five replicates of 10 individuals per replicate. They were fed with 0 (control), 0.25, 0.5 or 1 g rutin/kg supplementation in their basal diet. Dietary rutin supplementation, especially the 1 g/kg diet, increased body weight gain, the protein efficiency ratio (*p* < 0.001) and both white blood cell and lymphocyte counts (*p* < 0.001). However, it had no effect on total protein, albumin, globulin, or alanine transaminase. A high concentration of rutin (0.5 and 1 g/kg) also significantly reduced serum total cholesterol, triacylglycerol and low-density lipoprotein cholesterol concentrations (*p* < 0.001), as well as malondialdehyde concentrations (*p* = 0.001). A high concentration diet also increased the activity of superoxide dismutase, catalase and glutathione peroxidase. Of the lipid-related genes examined, acetyl CoA carboxylase and fatty acid synthase were significantly down-regulated in the livers of rutin-fed individuals, whereas carnitine palmitoyl transferase 1 and peroxisome proliferator-activated receptor alpha were significantly up-regulated. Therefore, rutin supplementation at 1 g/kg has the potential to improve the productive performance and health status of broiler chickens.

## 1. Introduction

The use of antibiotics as growth promoters has recently been prohibited all over the world, owing to increased concerns over food safety, environmental contamination and general health risks, which has subsequently spurred animal scientists and producers to identify alternative supplements that can be used to improve animal growth, immunity and meat quality [1]. For example, plant polyphenols have attracted attention as alternatives to antibiotics, owing to their growth-promoting potential, antioxidant capacity and immunomodulatory properties [2,3], and such polyphenols, especially flavonoids, have been reported to possess antimicrobial, anti-inflammatory, anti-carcinogenic, antithrombotic, antimutagenic and hepatoprotective properties [4]. Studies have investigated the effects of many flavonoids, including quercetin, rutin, hesperetin, naringenin and genistein, however, rutin has received a disproportionate amount of attention, owing to its biological importance.

Rutin (i.e., rutoside, vitamin P, quercetin-3-rutinoside or sophorin) is a flavonol glycoside that includes both flavonolic aglycone quercetin and disaccharide rutinose [5] and is found in certain plant products, including apples, citrus fruits, berries, black teas, passion flower and buckwheat. The compound exhibits a number of pharmacological activities, including antioxidant [6], anti-inflammatory [7], anticarcinogenic [8], neuroprotective [9], vasoprotective [10] and cardioprotective activities [11]. It also exhibits mucus protective and anti-ulcer effects by inhibiting the gastric proton pump [12]. Rutin can also be used to inhibit free radical-mediated cytotoxicity and lipid peroxidation, as well as preventing the oxidation of vitamin C to enhance its absorption. Therefore, rutin could possibly be used to prevent certain diseases and to protect genome stability, thereby improving animal production.

Therefore, the aim of the present study was to investigate the effects of rutin on the performance, blood biochemical biomarkers, antioxidant capacity and gene expression of broilers.

## 2. Materials and Methods

The experimental protocol was carried out in accordance with the regulations of the Local Experimental Animals Care Committee of the Faculty of Veterinary Medicine, Zagazig University, and was approved by the institutional ethics committee (Approval No. ANWD 215).

### 2.1. Birds, Diets and Management

One-day-old female Ross-308 chicks (45.15 ± 1.33 g, *n* = 200) were obtained from a local commercial hatchery and raised on floor pens with wood shavings as litter (bird density: 10 broilers/m^2^). Except for day 1, a 23 h:1 h light to dark cycle was used. The environmental temperature was maintained at about 32 °C during the first week and then gradually decreased by 2 °C each week until it reached 22 °C at the beginning of the sixth week. All diets were provided in mash form and the basal diets were formulated (starter, grower and finisher; Table 1) according to the recommendations of the Ross Broiler Pocket Guide, published in 2015 by Aviagen. Chemical analyses (moisture, crude protein and ether extract) of the feedstuffs and experimental diets were conducted according to the official methods of Reference [13]. Feed and water were provided ad libitum. All chicks were vaccinated against the Gumboro and Newcastle diseases, according to the manufacturer’s (Intervet Inc. Company, Millsboro, DE, USA) instructions.

### 2.2. Experimental Design

The broiler chicks were randomly distributed among four experimental treatments, with five replicates of 10 individuals per treatment. Each replicate was assigned to a clean floor pen (1 m^2^) and fed a basal diet supplemented with 0 (control), 0.25, 0.5 or 1 g rutin powder/kg. The rutin (C27H30O16) was obtained from Sigma-Aldrich (St. Louis, MO, USA).

### 2.3. Performance Traits

All chicks were weighed individually on day 1 (initial body weight) and day 42 (final body weight), and body weight gain was calculated as the difference between the final body weight and the initial body weight. The feed was removed 2 h before weighing the birds. Feed intake during the period was calculated for each replicate on day 42 and the feed conversion rate was calculated for each replicate as feed intake divided by mean body weight gain [14]. The protein efficiency ratio was calculated according to Reference [15], as the weight gain (g) produced per unit of dietary protein consumed. The relative growth rate (RGR) was calculated as described by [16]:RGR = (final weight − initial weight)/[0.5 × (final weight + initial weight) × 100](1)

### 2.4. Sample Collection

At the end of the experimental period, ten birds were randomly selected from each group and fasted for 12 h. Blood samples (*n* = 40) were collected from the brachial vein of the birds, with one part transferred to a gel activator tube for serum separation (3000 rpm; 15 min; 4 °C) and the other part transferred to a vacutainer tube that contained ethylenediaminetetraacetic acid (EDTA) as an anticoagulant for hematological analysis. After blood sampling, the chickens (*n* = 40) were euthanized, manually feathered and eviscerated. Liver samples were rapidly excised, flushed with ice-cold phosphate-buffered saline and snap-frozen in liquid nitrogen for subsequent RNA isolation. 

### 2.5. Hematological Evaluation

The total red blood cells, hematocrit, hemoglobin, mean cell volume, mean cell hemoglobin, mean corpuscular hemoglobin concentration, total leukocytes, lymphocytes and heterophils of the whole blood samples were measured using a Hema Screen 18 automated hematology analyzer (Hospitex Diagnostics, Sesto Fiorentino, Italy) as described previously [17]. The total leukocyte count was measured using an automated analyzer, whereas differential leukocyte counts were calculated manually [18].

### 2.6. Serum Biochemical Indices and Antioxidant Capacity

Serum total protein, albumin, triacylglycerol, total cholesterol, high-density lipoprotein cholesterol, low-density lipoprotein cholesterol, alanine aminotransferase and aspartate aminotransferase were analyzed using a commercially available diagnostic kit (Spinreact Co., Santa Coloma, Spain). Serum concentrations of malonaldehyde, superoxide dismutase (SOD), catalase (CAT) and glutathione peroxidase (GSH-Px) were measured using other commercially available assay kits (Institute Co., Nanjing, Jiangsu, China), according to References [19,20,21,22], respectively.

### 2.7. Antioxidant and Lipid-Related Gene Expression in the Liver by RQ-PCR

The total RNA was extracted from the liver tissue samples using the QIAamp RNeasy Mini Kit (Qiagen GmbH, Düsseldorf, Germany) following the manufacturer’s protocol. The concentration and purity of the resulting RNA were estimated using a GeneQuant spectrophotometer (Pharmacia Biotech, Freiburg, Germany). The total RNA was then reverse transcribed into complementary DNA (cDNA) using the RevertAid Reverse Transcription Kit (Thermo Fisher, Waltham, MA, USA) according to the manufacturer’s instructions. Real-time PCR analysis was performed using the QuantiTect SYBR Green PCR Kit (Qiagen GmbH, Düsseldorf, Germany), where β-actin was used as the internal control. Chicken β-actin, *ACC*, *FAS*, *CPT1*, *PPARα*, *SOD*, *CAT* and *GSH-PX* were amplified using gene-specific primer sequences (Table 2) and the qRT-PCR was performed using 25 µL reactions that contained 12.5 µL of 2X QuantiTect SYBR Green PCR Master Mix (Qiagen GmbH, Germany), 0.25 µL of RevertAid Reverse Transcriptase (200 U/µL; Thermo Fisher, Waltham, MA, USA), 0.5 µL of each primer (20 pmol), 8.25 µL of water and a 3 µL cDNA template. This was performed with the following cycling conditions: 15 min at 95 °C, followed by 40 cycles of 15 s at 95 °C, 15 s at 60 °C and 15 s at 72 °C. Melt-curve analysis was performed (from 65 to 95 °C, using 0.5 °C temperature increments with a 5 s hold at each step) using a Stratagene MX3005P (Thermo Fisher, Waltham, MA, USA). The relative fold change in the expression of target genes was calculated using the comparative 2^−ΔΔCt^ method [23], where ΔΔCt indicates the difference between the mean ΔCt of the treatment group and that of the control group, and ΔCt represents the difference between the mean Ct of the gene of interest and that of the internal control gene for each sample.

### 2.8. Economic Efficiency of Diet

The economic viability of rutin supplementation was calculated from the input-output analysis (as per the prevailing market price of the experimental diets and the broiler live body weight at the time of the experiment) as follows:Total feed cost = total feed intake per bird × cost of 1 kg diet(2)
Feed cost/kg weight gain = feed conversion × cost of 1 kg diet(3)
Profit/kg weight gain = return/kg gain (price/kg meat) − feeding cost/kg gain(4)
Benefit-cost ratio = profit/feed cost per kg gain(5)

### 2.9. Statistical Analysis

All statistical procedures were performed using SPSS v. 23.0 (SPSS Inc., Chicago, IL, USA), with replicates treated as experimental units. The data were analyzed using the general linear model (GLM) procedure of SPSS (a one-way analysis of variance), after verifying normality using Shapiro-Wilk’s test and the homogeneity of the variance components between experimental groups using Levene’s test. Tukey’s honestly significant difference (HSD) multiple comparison test was used to test for significant differences between the mean values. Variation in the data was expressed as the pooled standard error of the mean (SEM) and the alpha level for determination of significance was set at 0.05.

## 3. Results

### 3.1. Effect of Dietary Rutin on Performance Traits

Broilers fed a diet supplemented with the highest level of rutin (1 g/kg) exhibited a greater body weight, body weight gain, protein efficiency ratio and a lower feed conversion ratio (FCR) (*p* < 0.001) compared to the other groups (Table 3). However, they exhibited no linear or quadratic effect on either feed intake or the relative growth rate.

### 3.2. Effect of Dietary Rutin on Hematological Parameters

The effect of dietary rutin on the blood-based parameters of broilers is listed in Table 4. The higher levels of rutin supplementation (0.5 and 1 g/kg) linearly increased white blood cell (WBC) and lymphocyte counts (*p* < 0.001) but had no effect on the other blood constituents.

### 3.3. Effect of Dietary Rutin on Serum Biochemical Indices and Antioxidant Capacity

Data regarding serum biochemicals and antioxidant capacity are presented in Table 5. Rutin supplementation failed to affect protein, albumin and globulin levels or the concentration of high-density lipoprotein cholesterol, however it did reduce the total cholesterol, triacylglycerol and low-density lipoprotein cholesterol levels, especially for the 0.5 and 1 g/kg diets (*p* < 0.001). The highest level of rutin supplementation (1 g/kg) reduced the alanine aminotransferase level. Furthermore, the higher levels of rutin supplementation (0.5 and 1 g/kg) linearly increased SOD, CAT, and GSH-Px activity and reduced malonaldehyde concentrations (*p* = 0.001) but had no effect on the concentration of aspartate aminotransferase (Table 5).

### 3.4. Effect of Dietary Rutin on Antioxidant and Lipid-Related Gene Expression

The expression of *SOD*, *CAT* and *GSH-PX* was higher in birds fed the highest level of rutin (1 g/kg; Figure 1), whereas dose-dependent down-regulation was observed for the expression of hepatic lipid anabolism genes (*ACC*, *FAS*), with more pronounced effects in the birds supplemented with higher levels of rutin (0.5 and 1 g/kg). There was a non-significant increase in those fed with the lowest level of rutin (0.25 g/kg; Figure 2). Meanwhile, the expression of fat transportation and catabolism-related genes (*CPT1* and *PPAR-α*) was up-regulated in the birds supplemented with higher levels of rutin (0.5 and 1 g/kg), with a similar but non-significant increase observed in birds fed the lowest level of rutin (0.25 g/kg).

### 3.5. Effect of Dietary Rutin on Economic Efficiency

The diet supplemented with the highest level of rutin (1 g/kg) had the highest total feed cost among the treatment groups (*p* < 0.001, Table 6). However, feed cost/kg gain was unaffected by rutin supplementation, and even though rutin supplementation generally improved final body weight, the profit and benefit-cost ratio were unaffected.

## 4. Discussion

The present study concludes that rutin supplementation is capable of improving the body weight, body weight gain and FCR of broiler chickens, likely because of the favorable effect of flavonoids on gut morphology and the functional architecture of the small intestine [24,25]. It is also possible that flavones up-regulate the combination of the growth hormone and the hepatic growth hormone receptor, which increases insulin-like growth factor 1 concentrations, thereby promoting animal growth [26]. In general, dietary flavonoid supplementation has been reported to improve the growth performance of broiler chickens [27,28]. For example, both Koehler (2002) and Islam et al. (2016) reported that buckwheat, a rich source of rutin, significantly improved the final body weight and FCR of broilers [29,30]. However, it is important to note that Sayed et al. (2015) reported that buckwheat had no effect on either growth or feed intake [31].

Consistent with previous reports [32,33], the present study found that rutin supplementation had no effect on the total number of red blood cells, hematocrit, hemoglobin, mean cell volume, mean cell hemoglobin or mean corpuscular hemoglobin concentration. The study also found that higher levels of rutin supplementation (0.5 and 1 g/kg) improved both WBC and lymphocyte counts but had no effect on the heterophil count or the heterophil/lymphocyte (H/L) ratio, thereby demonstrating an improvement in the non-specific immune response of the broilers. It is important to note that rutin supplementation failed to induce any significant changes in either the WBC or lymphocyte counts of hamsters [34]. However, this disparity may be a result of the difference in species, the hygienic status of the study location, environmental conditions or diet composition [35].

Furthermore, rutin supplementation had no effect on serum protein, albumin, or globulin levels. This is in agreement with a previous study on isoflavone sources [36]. However, the present study observed a dose-dependent decrease in serum cholesterol, triacylglycerol and low-density lipoprotein cholesterol levels, which suggests that rutin plays an important role in modulating lipid metabolization. The dramatic reduction in cholesterol indicates that rutin inhibits 3-hydroxy-3-methyl-glutaryl-coenzyme A (HMG-CoA) reductase, a key enzyme in cholesterol biosynthesis [37]. Indeed, previous studies have suggested that rutin reduces cholesterol synthesis by reducing the amount of free fatty acids available for triacylglycerol synthesis through the stimulation of *PPAR-α* and downstream target enzymes [38]. These studies also reported that dietary flavonoid supplementation had similar positive effects on plasma cholesterol and triacylglycerol levels [36,39]. Notably, Chuffa et al. (2014) reported that rutin injections prompted notably better lipid profiles in the livers, hearts and blood of mice fed a high-calorie, alcohol-containing diet [40]. Meanwhile, Cavallini et al. (2009) reported that isoflavones failed to reduce cholesterol levels in hypercholesterolemic rabbits [41]. However, this may be a result of the difference in species, especially considering that differences in gut microflora can affect flavonoid metabolism [42].

The present study demonstrated that dietary rutin supplementation has a significant effect on serum alanine aminotransferase levels, which suggests that rutin possesses hepatoprotective properties and that these properties could possibly be attributed to cytokine production, which has been reported to provide hepatoprotection in a variety of liver injury models [43]. Al-Rejaie et al. (2013) reported that rutin supplementation provided protective effects against hepatotoxicity in male Wistar rats fed a high-cholesterol diet, as indicated by reduced plasma levels of alanine transaminase, aspartate aminotransferase, triacylglycerol, total cholesterol and low-density lipoprotein [44], and Janbaz et al. (2002) reported that rutin exhibited a hepatoprotective activity against paracetamol and carbon tetrachloride (CCL4) in rodents by restoring aspartate aminotransferase and alanine aminotransferase levels [45].

The free radicals and oxidative stress processes that occur in living organisms have long attracted the attention of scientific research. However, the current emphasis of such research has shifted towards the investigation of natural antioxidants, such as flavonoids, which are extremely important in maintaining health and preventing disease. Among different flavonoid compounds, rutin is a potent antioxidant. Rutin exerts its antioxidant effects by scavenging free radicals through inhibiting xanthine oxidase activity and lipid peroxidation and by chelating iron [46], and the antiradical activity of rutin can be mainly attributed to free hydroxyl groups on C4’, C3’, and C7 [47]. In the present study, rutin supplementation markedly enhanced the activities of GSH-PX, SOD, and CAT but reduced serum malonaldehyde levels in a dose-dependent manner, thereby suggesting that rutin is able to transfer electrons and free radicals, in addition to its ability to activate antioxidant enzymes and reduce oxidative stress [48]. This is in agreement with previous reports that flavonoids improve the antioxidant status of broilers [49,50] and that dietary bioflavonoid supplementation improves the anti-oxidative status and free radical scavenging activity in layer chickens [51], rabbits [52], turkeys [53] and mice [54]. In particular, the antioxidant activity of rutin has been studied in various model systems [55,56,57,58].

In addition to demonstrating the positive effects of rutin on serum antioxidant capacity, the present study also provides evidence that dietary rutin supplementation may affect the expression of certain antioxidant-related genes. More specifically, rutin levels were positively associated with significant increases in *SOD*, *CAT* and *GSH-PX* expression in broiler livers, which is consistent with previous reports that plant polyphenols are effective antioxidants and improve the systemic anti-oxidative status and gene expression of animals [39,44,59].

Because lipid metabolism is regulated by a variety of genes that are related to anabolism and catabolism [60], the present study evaluated the expression of several factors known for their importance in lipid anabolism (*ACC* and *FAS*) and lipid catabolism (*CPT1* and *PPAR-α*) in the livers of broiler chickens. The results revealed that rutin supplementation, especially at 1 g/kg, down-regulated the expression of both *ACC*, which is a rate-limiting enzyme of fatty acid synthesis and catalyzes acetyl-CoA to generate malonyl-CoA [61] and FAS, which regulates the rate-limiting step in fatty acid synthesis. Indeed, *ACC* and *FAS* are highly correlated to lipogenesis [62], and previous studies have also reported that polyphenols inhibit the activity and expression of *FAS*, which functions as a competitive inhibitor of NADPH (a substrate of b-ketoacyl reductase of type I and II FAS), thereby suggesting that polyphenols compete with NADPH for the same binding site [26,63]. On the other hand, *CPT1* and *PPAR-α*, which play important roles in the regulation of glucose, lipid metabolism and fatty acid oxidation [60,64], were significantly up-regulated, which indicates that fatty acid β-oxidation and triglyceride hydrolysis were higher in rutin-supplemented groups which were associated with a better gain-to-feed ratio and nutrient digestibility, as mentioned by Reference [65]. Pan et al. (2016) reported that polyphenols promote lipid metabolism by activating AMP-activated protein kinase, thereby reducing lipogenesis, enhancing lipolysis, and ultimately reducing lipid accumulation by inhibiting the differentiation and proliferation of preadipocytes [63].

## 5. Conclusions

The results of the present study suggest that supplementing broiler diets with rutin, especially at 1 g/kg, has a variety of growth-promoting effects, enhances antioxidant capacity and suppresses lipogenesis, thereby reducing fat deposition and serum lipid levels. The results also demonstrate that the observed benefits can be achieved without compromising economic profits. Further studies should be conducted to accurately assess and evaluate the efficiency of rutin supplementation against the impact of pathogens and other immunosuppressive stressors. 

## Figures and Tables

**Figure 1 animals-09-00007-f001:**
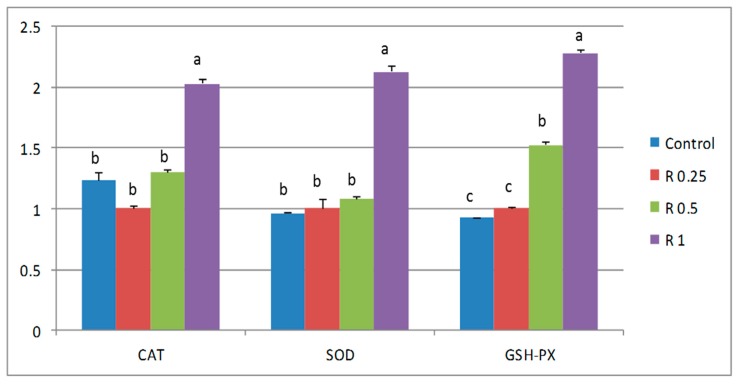
Effect of dietary rutin on the relative expression of *CAT*, *SOD* and *GSH-PX* in broiler livers. Each bar carrying different letters (a, b, c) was significantly different (*p* < 0.05) (mean ± standard error, *n* = 10).

**Figure 2 animals-09-00007-f002:**
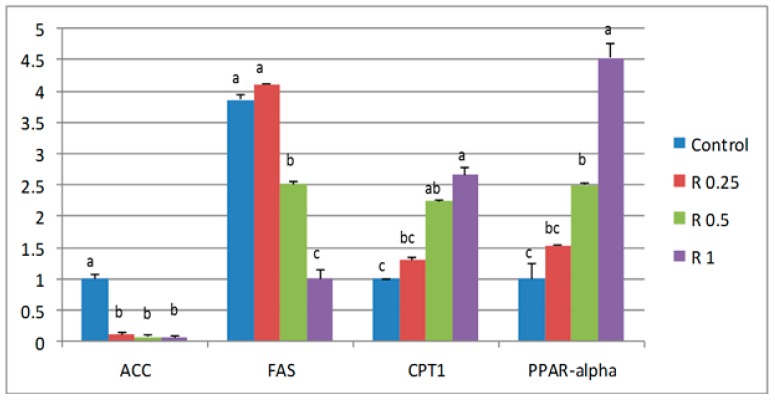
Effect of dietary rutin on the relative expression of *ACC*, *FAS*, *CPT1*, and *PPAR-α* in broiler livers. Each bar carrying different letters (a, b, c) was significantly different (*p* < 0.05) (mean ± standard error, *n* = 10).

**Table 1 animals-09-00007-t001:** Ingredients and chemical composition of basal diets (as-fed basis).

Ingredients	Starter (0 to 10 d)	Grower (11 to 24 d)	Finisher (25 to 42 d)
Yellow corn	56.00	60.60	62.00
Soybean meal, 48%	34.86	29.00	25.00
Corn gluten, 60%	3.50	4.50	4.00
Wheat bran	-	-	1.90
Soybean oil	1.80	2.00	3.66
Calcium carbonate	1.00	1.00	0.90
Dicalciumphosphate	1.80	1.90	1.60
Common salt	0.30	0.30	0.30
Premix *	0.30	0.30	0.30
DL- Methionine, 98%	0.18	0.14	0.11
Lysine, HCl, 78%	0.16	0.16	0.13
Toxenil	0.10	0.10	0.10
Calculated chemical composition ^†^			
ME, Kcal/Kg	3042.27	3104.52	3202.02
CP, %	23.30	21.44	19.57
EE, %	4.28	4.60	6.24
CF, %	2.64	2.55	2.63
Ca, %	0.97	0.98	0.86
Available P, %	0.47	0.48	0.41
Lysine, %	1.38	1.22	1.10
Methionine, %	0.57	0.52	0.46

* Supplied per kg of diet: Vitamin A, 12,000 IU; vitamin D3, 2200 IU; vitamin E, 26 IU; vitamin K3, 6.25 mg; vitamin B1, 3.75 mg; vitamin B2, 6.6 mg; vitamin B6, 1.5 g; pantothenic acid, 18.8 mg; vitamin B12, 0.31 mg; niacin, 30 mg; folic acid, 1.25 mg; biotin, 0.6 mg; Fe, 50 mg; Mn, 60 mg; Cu, 6 mg; I, 1 mg; Co, 1 mg; Se, 0.20 mg; Zn, 50 mg; and choline chloride, 500 mg. ^†^ Calculated according to NRC (1994) tables. ME: Metabolic energy; CP: Crude protein; EE: Ether extract; CF: Crude fiber; Ca: Calcium; P: Phosphorus.

**Table 2 animals-09-00007-t002:** Primer sequences used for qRT-PCR analysis.

Gene	Sequence (5′ → 3′)	GenBank Number
*ß-actin*	F: ATTGTCCACCGCAA ATGCTTC R: AAATAAAGCCATGCCAATCTCGTC	NM_205518.1
*GSH-PX*	F: TTGTAAACATCAGGGGCAAA R: ATGGGCCAAGATCTTTCTGTAA	NM_001163245.1
*SOD*	F: AGGGGGTCATCCACTTCC R: CCCATTTGTGTTGTCTCCAA	NM_205064.1
*CAT*	F: ACCAAGTACTGCAAGGCGAA R: TGAGGGTTCCTCTTCTGGCT	NM_001031215.1
*ACC*	F: AATGGCAGCTTTGGAGGTGT R: TCTGTTTGGGTGGGAGGTG	NM_205505
*FAS*	F: CTATCGACACAGCCTGCTCCT R: CAGAATGTTGACCCCTCCTACC	J03860
*CPT1*	F: CAATGAGGTACTCCCTGAAA R: CATTATTGGTCCACGCCCTC	AY675193
*PPARα*	F: TGGACGAATGCCAAGGTC R: GATTTCCTGCAGTAAAGGGTG	AF163809

*GSH-PX*: Glutathione peroxidase; *SOD*: Superoxide dismutase; *CAT*: Catalase; *ACC*: Acetyl CoA carboxylase; *FAS*: Fatty acid synthase; *CPT1*, carnitine palmitoyl transferase 1; *PPAR-α*, peroxisome proliferator-activated receptor alpha; F: Forward primer; R: Reverse primer.

**Table 3 animals-09-00007-t003:** Effect of dietary rutin supplementation on the performance traits of broiler chickens.

Parameters	Rutin, g/kg Base Diet				SEM	*p*-Value	
	0	0.25	0.5	1		Linear	Quadratic
Initial body weight, g	45.60	44.80	45.20	45.00	0.36	0.688	0.700
Final body weight, g	1953 ^c^	2012 ^b,c^	2079 ^b^	2225 ^a^	14.55	0.000	0.012
Body weight gain, g	1907 ^c^	1967 ^b,c^	2034 ^b^	2180 ^a^	15.58	0.000	0.013
Total feed intake, g	3611	3646	3685	3692	11.83	0.202	0.838
Feed conversion ratio	1.89 ^a^	1.85 ^a^	1.81 ^a^	1.69 ^b^	0.02	0.001	0.099
Protein efficiency ratio	2.59 ^b^	2.63 ^b^	2.61 ^b^	2.79 ^a^	0.11	0.000	0.051
Relative growth rate, %	190.83	191.28	191.51	192.07	2.16	0.061	0.702

^a,b,c^ Different superscripts within each row indicate significant differences (*p* < 0.05). SEM: Standard error of the mean.

**Table 4 animals-09-00007-t004:** Effect of dietary rutin on the hematological parameters of broiler chickens.

Parameters	Rutin, g/kg Base Diet				SEM	*p*-Value	
	0	0.25	0.5	1		Linear	Quadratic
RBCs, 10^6^/µL	2.45	2.46	2.51	2.52	0.05	0.584	0.933
Hb, g/dL	6.21	6.24	6.32	6.25	0.13	0.760	0.783
HCT, %	25.92	24.96	26.58	27.05	0.29	0.063	0.239
MCV, fl	132.66	134.39	133.64	134.15	1.73	0.241	0.375
MCH, pg	41.99	42.66	42.86	43.06	0.23	0.102	0.605
MCHC, g/dL	30.41	30.55	31. 08	31.36	0.29	0.190	0.929
WBCs, 10^3^/µL	19.35 ^b^	20.65 ^a,b^	21.88 ^a^	22.24 ^a^	0.26	0.000	0.254
Lymphocyte, 10^3^/µL	13.61 ^b^	14.78 ^a,b^	15.54 ^a^	15.81 ^a^	0.21	0.000	0.187
Heterophil, 10^3^/µL	4.78	5.08	5.64	5.57	0.14	0.080	0.468
H/L ratio	0.35	0.34	0.36	0.35	0.01	0.919	0.764

^a,b^ Different superscripts within each row indicate significant differences (*p* < 0.05). RBCs: Red blood cells; Hb: Hemoglobin; HCT: Hematocrit; MCV: Mean corpuscular volume; MCH: Mean corpuscular hemoglobin; MCHC: Mean corpuscular hemoglobin concentration; WBCs: White blood cells; H/L: Heterophil/lymphocyte; SEM: Standard error of the mean.

**Table 5 animals-09-00007-t005:** Effect of dietary rutin on serum biochemical and antioxidant parameters of broiler chickens.

Parameters	Rutin, g/kg Base Diet				SEM	*p*-Value	
	0	0.25	0.5	1		Linear	Quadratic
Serum indices							
Total protein, g/dL	5.98	5.99	6.11	6.60	0.12	0.063	0.275
Albumin, g/dL	3.23	3.23	3.42	3.62	0.09	0. 065	0.597
Globulin, g/dL	2.75	2.76	2.69	2.98	0.09	0.477	0.484
Cholesterol, mg/dL	131.58 ^a^	121.53 ^a,b^	112.26 ^b^	91.15 ^c^	2.79	0.000	0.046
Triacylglycerol, mg/dL	94.08 ^a^	91.06 ^a^	92.06 ^a^	78.14 ^b^	1.25	0.000	0.001
HDL cholesterol, mg/dL	41.62	40.14	39.15	37.16	1.62	0.067	0.944
LDL cholesterol, mg/dL	70.15 ^a^	60.91 ^a,b^	52.21 ^b^	36.23 ^c^	0.78	0.000	0.040
ALT, U/L	110.63 ^a^	109.21 ^a^	104.12 ^a^	87.12 ^b^	3.94	0.000	0.005
AST, U/L	75.11	75.76	73.63	70.12	4.49	0.246	0.536
Antioxidant parameters							
SOD, U/mL	17.12 ^c^	19.18 ^b,c^	21.22 ^ab^	24.37 ^a^	1.12	0.008	0.460
CAT, U/mL	9.12 ^b^	11.18 ^b^	15.31 ^a^	15.91 ^a^	1.48	0.000	0.209
GSH-PX, U/mL	2.10 ^b^	3.21 ^a^	3.42 ^a^	3.57 ^a^	0.13	0.000	0.014
MDA, nmol/mL	3.87 ^a^	2.48 ^b^	1.18 ^c^	1.11 ^c^	0.04	0.001	0.008

^a,b,c^ Means bearing different superscripts within the same row are significantly different (*p* < 0.05). HDL cholesterol: High density lipoprotein-cholesterol; LDL cholesterol: Low density lipoprotein-cholesterol; ALT: Alanine transaminase; AST: Aspartate aminotransferase; SOD: Superoxide dismutase; CAT: Catalase; GSH-PX: Glutathione peroxidase; MDA: Malondialdehyde; SEM: Standard error of the mean.

**Table 6 animals-09-00007-t006:** Effect of dietary rutin on the economic efficiency of broilers (at 42 days of age).

Parameters	Rutin, g/kg Control Diet				SEM	*p*-Value	
	0	0.25	0.5	1		Linear	Quadratic
Total feed cost/bird, $	1.62 ^d^	1.71 ^c^	1.81 ^b^	1.99 ^a^	0.03	0.000	0.021
Feed cost/kg gain, $	0.85	0.87	0.89	0.91	0.08	0.090	0.939
Profit/kg gain, $	0.62	0.60	0.58	0.56	0.07	0.075	0.941
Benefit-cost ratio	0.73	0.69	0.65	0.62	0.16	0.081	0.997

^a,b,c,d^ Different superscripts within each row indicate significant differences (*p* < 0.05). SEM: Standard error of mean. Cost/kg diet (including herbal cost) = $0.45 for control, $0.47 for 0.25 g/kg rutin, $0.49 for 0.5 g/kg rutin, $0.54 for 1 g/kg rutin; price/kg meat = $1.47.

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
