# Peer review of "Growth Performance, Antioxidant Capacity, Lipid-Related Transcript Expression and the Economics of Broiler Chickens Fed Different Levels of Rutin"

_animals, 2018, doi:10.3390/ani9010007_

Round 1
Reviewer 1 Report
The paper provide interesting insight on the use of dietary rutin supplementation.
Please provide the number of slaughtered birds and blood samples that have been collected (in paragraph 2.4 line 104).
In order to improve the scientific soundness I suggest to improve the statistic analysis method:
because of you are using increasing levele (linear increase) of the same supplement you should better study the Linear and Quadratic response by the polinomial contrast.
Furthermore the normality or non-normality of data distribution should be testd before statistics by Shapiro-Wilk's test.
Author Response
The paper provides interesting insight on the use of dietary rutin supplementation.
R: We thank the reviewer for the excellent reviewing and the constructive comments and suggestions to improve the quality of the manuscript.
Q1- Please provide the number of slaughtered birds and blood samples that have been collected (in paragraph 2.4 line 104).
R1- We added the numbers.
Q2- In order to improve the scientific soundness I suggest to improve the statistic analysis method: because of you are using increasing levele (linear increase) of the same supplement you should better study the Linear and Quadratic response by the polinomial contrast. Furthermore, the normality or non-normality of data distribution should be testd before statistics by Shapiro-Wilk's test.
R2- We thank the reviewer for this note. We followed the reviewer’s suggestion and reanalyzed the data and represented the results in the edited tables.
Reviewer 2 Report
This is an interesting paper and useful findings have been reported. The topic fits well withis the overall scope of Animals journal. In my opinion the paper metits the acceptance after revision (as I suggested in the attached PDF full-text file). In particular, the manuscript needs a revision for English language used and the Authors have to pay attention to the references style when cited in the text. Further, some references cited in the text have been missed in the list.
Here are some examples:
Line 27: 42 day → 42 days;
Line 27: 200 → A total of 200;
Line 29: on basal diet → in basal diet
Lines 45-48: Add this reference: Jahromi MF, Altaher YW, Shokryazdan P, Ebrahimi R, Ebrahimi M, Idrus Z, Goh YM, Tufarelli V, Liang JB (2016) Dietary supplementation of a mixture of lactobacillus strains enhances performance of broiler chickens raised under heat stress conditions. Int J Biometeorol 60:1099–1110.
Line 50: reference 1. Add also: Tufarelli, V., Casalino, E., D'Alessandro, A. G., & Laudadio, V. (2017). Dietary phenolic compounds: biochemistry, metabolism and significance in animal and human health. Current Drug Metabolism, 18(10), 905-913.
Line 66: Accordingly → Therefore
Please follow the journal's style for references, cite the name in the sentence and cite the number in the end of sentences, such as: lines 230, 242, reference [32], [35], [36], [53], [55]...
Lines 257-260: Please revise.
Line 276: This reference was missed in the references list. Please add, following the journal's style.
Author Response
This is an interesting paper and useful findings have been reported. The topic fits well withis the overall scope of Animals journal. In my opinion the paper metits the acceptance after revision (as I suggested in the attached PDF full-text file). In particular, the manuscript needs a revision for English language used and the Authors have to pay attention to the references style when cited in the text. Further, some references cited in the text have been missed in the list.
R: We thank the reviewer for the excellent reviewing and the constructive comments and suggestions to improve the quality of the manuscript. We have asked a professional English scholar to proofread the manuscript and to improve the language and English writing. We hope that the edited manuscript meets the quality of publishing in “Animals”.
Here are some examples:
Q1- Line 27: 42 day → 42 days;
R1- It has been Changed.
Q2- Line 27: 200 → A total of 200;
R2- It has been added.
Q3- Line 29: on basal diet → in basal diet
R3- R1- It has been Changed.
Q4- Lines 45-48: Add this reference: Jahromi MF, Altaher YW, Shokryazdan P, Ebrahimi R, Ebrahimi M, Idrus Z, Goh YM, Tufarelli V, Liang JB (2016) Dietary supplementation of a mixture of lactobacillus strains enhances performance of broiler chickens raised under heat stress conditions. Int J Biometeorol 60:1099–1110.
R4- R2- It has been added.
Q5- Line 50: reference 1. Add also: Tufarelli, V., Casalino, E., D'Alessandro, A. G., & Laudadio, V. (2017). Dietary phenolic compounds: biochemistry, metabolism and significance in animal and human health. Current Drug Metabolism, 18(10), 905-913.
R5- R2- It has been added.
Q6- Line 66: Accordingly → Therefore
R6- R1- It has been Changed.
Q7- Please follow the journal's style for references, cite the name in the sentence and cite the number in the end of sentences, such as: lines 230, 242, reference [32], [35], [36], [53], [55]...
R7- We thank the reviewer for this note. We followed the suggestion and changed them accordingly.
Q8- Lines 257-260: Please revise.
R8- It has been revised accordingly.
Q9- Line 276: This reference was missed in the references list. Please add, following the journal's style.
R9- We apologize for that. We added it.